# Inside Current Winemaking Challenges: Exploiting the Potential of Conventional and Unconventional Yeasts

**DOI:** 10.3390/microorganisms11051338

**Published:** 2023-05-19

**Authors:** Nunzio A. Fazio, Nunziatina Russo, Paola Foti, Alessandra Pino, Cinzia Caggia, Cinzia L. Randazzo

**Affiliations:** 1Department of Agriculture, Food and Environment, University of Catania, Via S. Sofia, 100, 95123 Catania, Italy; 2ProBioEtna Srl, Spin off University of Catania, Via S. Sofia 100, 95123 Catania, Italy

**Keywords:** wine, *Saccharomyces cerevisiae*, non-*Saccharomyces*, fermentation, sustainability, ethanol reduction, wine aroma, functional wine, starter culture

## Abstract

Wine represents a complex matrix in which microbial interactions can strongly impact the quality of the final product. Numerous studies have focused on optimizing microbial approaches for addressing new challenges to enhance quality, typicity, and food safety. However, few studies have investigated yeasts of different genera as resources for obtaining wines with new, specific traits. Currently, based on the continuous changes in consumer demand, yeast selection within conventional *Saccharomyces cerevisiae* and unconventional non-*Saccharomyces* yeasts represents a suitable opportunity. Wine fermentation driven by indigenous yeasts, in the various stages, has achieved promising results in producing wines with desired characteristics, such as a reduced content of ethanol, SO_2_, and toxins, as well as an increased aromatic complexity. Therefore, the increasing interest in organic, biodynamic, natural, or clean wine represents a new challenge for the wine sector. This review aims at exploring the main features of different oenological yeasts to obtain wines reflecting the needs of current consumers in a sustainability context, providing an overview, and pointing out the role of microorganisms as valuable sources and biological approaches to explore potential and future research opportunities.

## 1. Introduction

Wine is one of the earliest known fermented beverages, and it is strongly linked to a region and its people’s culture and customers. The needs of customers, even more concerned with climate change, sustainability, and health, have changed the way wine is produced in recent decades.

The world’s top wine producer is the European Union (Figure 1), with an estimated annual supply between 2016 and 2020 of 165 million hectoliters [1]. It accounted for 48% of consumption, 64% of output, and for 45% of wine-growing regions around the globe in 2020. Wine provided 7.6% of the value of all agri-food exports in the EU in 2020 [2]. From 2017 to 2022 (forecast updated to 31 October 2022), Italy is forecast to rank first with an average production of 48.8 Mio hl, followed by France, with 42.4 Million hL, and Spain, with 37.4 Million hL [3]. The wine industry has changed over time as consumers and producers have become more conscious of “green” issues pertaining to wine production, such as sustainability and carbon footprint. To satisfy niche market demands, many different wine subsets have been produced, such as organic, biodynamic, natural, or clean wine [4,5,6]. Moreover, starting in the 1990s, the Lifestyle of Health and Sustainability (LOHAS), a new way of living that prioritizes sustainability and health, has started to catch on all over the world. Customers show high levels of environmental consciousness, and one of the main objectives is to value quality over quantity. According to this philosophy of life, such consumers are willing to pay a superior price for intangible product qualities such as respect for environmental quality, human rights, and health [7]. In recent years, this mindset has had a substantial impact on consumer decisions in the agri-food industry, including the wine sector [8]. There is a strong environmental connotation in this context given the notable increase in consumer interest in purchasing wines from sustainable production on a global scale [9,10]. According to this trend, since 2013, organic wine consumption has doubled (976 million bottles produced) in contrast to traditional wine, which is seeing a decrease; organic wine is also currently one of the most important parts of the Italian food sector, with its 107,143 hectares of wine growing surface (+102% in 2019 vs. 2010) [11,12].

Alcoholic fermentation (AF) and, more broadly, the biochemical or chemical transformation of must into wine are processes carried out by yeasts, which are mainly responsible for the final complex quality of wine [13]. The main two groups of oenological yeasts involved are the *Saccharomyces* genus, particularly *Saccharomyces cerevisiae*, with a key role in must AF, and non-*Saccharomyces* (NS) yeasts, a heterogeneous group that includes autochthonous strains, therefore related to a particular area, and which have recently been studied for their positive effects on wine fermentation [14,15].

Currently, the increasing shift by consumers toward organic wines, perceived as more natural and healthier than traditional wines, has encouraged the bioprospecting of indigenous microorganisms (naturally occurring yeasts and bacteria) [16]. This change in consumer perception, as schematized in Figure 2, induces producers to select yeasts with distinctive technological traits. Strains of NS, therefore, can contribute to the satisfaction of customer needs by being able to: reduce production of SO_2_, acetaldehyde, H_2_S, and ethanol; control spontaneous and spoilage microbiota growth [17]; and improve the nutritional value or reduce detrimental molecule production, such as biogenic amines (BAs), mycotoxins (OTAs), and ethyl carbamate (EC) [18].

The use of commercial starters of *S. cerevisiae* strains makes the production safer, simpler, and more stable from a microbiological point of view and consequently avoids economic losses at the expense of originality and typicality [19]. Nonetheless, the combination with native NS strains might boost wines with higher sensorial complexity than those produced using solely *S. cerevisiae* [20], conferring a distinctive volatile imprint of a particular geographical location, enhancing the production of healthy-value compounds, improving primary and secondary wine aromas, color stabilization, reducing ethanol content, and controlling microbial spoilage (by releasing killer toxins) [21,22,23,24].

Additionally, to obtain consciousness of interactions among microorganisms, omics approaches have been successfully applied at different fermentation stages. In detail, metagenomic, proteomic, transcriptomic, and metabolomic studies are effective in revealing the most technologically useful yeasts, although several limitations, such as cost, large amounts of data, the design of experiments, a lack of complete knowledge of genes, and the quality of the samples to be analyzed, still remain [25].

According to the current scientific evidence, it becomes even clearer how important it is to investigate in depth the dynamics of the involved microbial population. In the present study, a literature survey was carried out, taking into account a fixed timeline between 2010 and 2022, using the keyword “wine” and searching Science Direct. In this way, 45,089 publications were found, 3206 of which related to biotechnology and applied microbiology fields. Zooming in on the “wine sustainability” category, in the same years, 1064 publications were found, 192 of which regarded food science technology. Nevertheless, as reported in Figure 3, it is also noteworthy to observe the increase in the number of published scientific papers throughout the years, indicating the ever-increasing relevance of the sustainability topic [26].

This review aims to report on the main challenges facing the wine industry, considering the increasing needs of consumers in terms of sustainability, health safety, and quality. In this context, the use of selected *Saccharomyces* and NS yeast strains in the production process represents a promising resource to meet the demands of both producers and consumers. The most commonly used oenological yeasts will be here described, taking into account both positive and negative traits, highlighting future perspectives in such an evolving sector, and pointing out the biotechnological progress in this field.

## 2. Microbiota of Vineyard and Fermenting Must

The microbiome presents in vineyards and involved in subsequent fermentation represents a complex and diverse system that includes the presence of different yeast species. In detail, the microbiota of grapes must originate mainly from grape skins and the winery environment. Yeasts are spatially distributed over grape berries and grape bunches, and their ecology varies according to the state of ripeness, as mature grapes present higher nutrient content, such as sugars [26,27]. As a matter of fact, at harvesting time, yeast populations on mature grapes are higher (10^3^ to 10^6^ CFU/g) than on immature grapes (10 to 10^3^ CFU/g) [28].

As a microbial reservoir, soil may affect wine’s color, fragrance, taste, and quality, as well as crop production and metabolite synthesis. Gammaproteobacteria, which include *Pseudomonas* species, and Firmicutes, which include *Bacillus* spp., are the taxa mainly found in soil; additionally, the same *S. cerevisiae* genotypes have been found to be shared between soil and fruit niches [29]. As extensively reported, a plethora of yeasts have been isolated from many grape varieties [30], mainly affiliated with the following genera: *Aureobasidium, Auriculibuller, Brettanomyces, Bulleromyces, Candida, Cryptococcus, Debaryomyces, Hanseniaspora, Issatchenkia, Kluyveromyces, Lipomyces, Metschnikowia, Pichia, Rhodosporidium, Rhodotorula, Saccharomyces, Sporidiobolus, Sporobolomyces, Torulaspora, Yarrowia, Zygoascus,* and *Zygosaccharomyces.* From an ecological perspective, apiculate yeasts, such as *Hanseniaspora* spp., abound on grape surfaces and represent over 50% of the yeast population, while yeasts of the *Candid*a, *Kluyveromyces*, *Metschnikowia*, *Pichia*, *Starmerella,* and *Cryptococcus* genera are found at lower densities [31]. Despite the extremely low occurrence of fermentative species of *Saccharomyce*s on the surface of unharmed berries or even in vineyard soils, *S. cerevisiae* is the unquestioned dominating species in the winemaking fermentation process. In addition, it is well documented that the dynamics of the yeast population during wine fermentation follow a predictable pattern of growth, with NS yeasts initially present at higher concentrations (about 10^4^ cells/mL) growing during the early stages (up to 4% *v/v* of ethanol), to be quickly replaced by strong fermentative *S. cerevisiae* strains [32]. In particular, during the initial stage of fermentation, microbiota is dominated by naturally occurring species with low ethanol tolerance, such as *Hanseniaspora*, *Pichia*, *Metschnikowia*, and *Candida*, as well as *Torulaspora*, *Lachancea*, *Saccharomycodes*, *Zygosaccharomyces*, *Issatchenkia*, *Meyerozyma*, *Wickerhamomyces*, *Filobasidium*, *Sporobolomyces*, *Rhodosporidium*, and *Vishniacozyma*, which are detected at lower densities. The diversity of the wine microbiota rapidly changes close to the conclusion of the fermentation process, with ethanol-sensitive yeasts, bacteria, and fungi largely replaced by *S. cerevisiae* [33]. To better understand the ecological background driving wine fermentation, it is relevant to predict the effects of environmental conditions on the metabolic pathway of yeasts that can switch from aerobic respiration to alcoholic fermentation. Under low O_2_ concentrations, yeasts conduct alcoholic fermentation as pyruvate is converted into ethanol and CO_2_, whereas at higher O_2_ concentrations, aerobic fermentation occurs and pyruvate is converted to acetyl CoA and CO_2_, with lower glucose consumption. Such a phenomenon, known as the “Pasteur effect”, is mainly reported for NS yeasts that prefer oxidative metabolism. In detail, this process is regulated by both pyruvate dehydrogenase (involved in respiration) and pyruvate decarboxylase (involved in fermentation) towards pyruvic acid in aerobic conditions and by ATP-mediated inhibition of the phosphofructokinase. In contrast, *S. cerevisiae*, in the presence of high sugar content (about 200 g/L), regardless of O_2_ presence, can drive alcoholic fermentation (a condition known as the “Crabtree effect”), which should be considered in the design of wine starters [34].

The application of PCR-based techniques, such as Single-Strand Conformational Polymorphism (SSCP), Denaturing Gradient Gel Electrophoresis (DGGE), Terminal Restriction Fragment Length Polymorphism (T-RFLP), and Automated rRNA Intergenic Spacer Analysis (ARISA), has upgraded the approach to studying wine microbiota, allowing to reveal microorganisms based on the presence of nucleic acids rather than their capacity to grow on specific media. The application of such techniques provides for the extraction of DNA to be used as a template for PCR directly from matrices, and it is generally quicker, more sensitive, and more accurate than culture-dependent techniques [35].

### 2.1. Saccharomyces cerevisiae

*S. cerevisiae* is the main yeast involved in wine fermentation and the most commonly employed microorganism for bread and beer production. Moreover, as a result of strong domestication, winemakers can select the best-performing *S. cerevisiae* strains to improve wine production in terms of flavor and fragrance from an enormous diversity of strains [36].

As previously reported, *S. cerevisiae* may be primarily found in soil and grape berries and at the initial stage of fermentation; however, it is not completely ruled out if it may be found on winery machinery, such as stemmer-crushers, pumps, pipes, or fermentation vessels, where it could contribute to enhancing biodiversity ecology [37].

From an enological point of view, *S. cerevisiae* is important for several traits, among which are tolerance to high ethanol concentrations and the production of killer factors.

As previously mentioned, thanks to the Crabtree effect, even with aerobic conditions, *S. cerevisiae* does not employ its respiratory system to metabolize saccharides and increase biomass growth, but instead, through pyruvate, it releases ethanol and other two-carbon molecules. As a result of this, *S. cerevisiae* creates and accumulates ethanol, which is deadly to most other microbial species capable of competing with it for sugar molecules and hence eliminates competition. After that, it continues to consume the produced ethanol, thus improving its own growth [38].

Focusing on must fermentation, higher alcohols (fusel alcohols), from a quantitative point of view, are the most important compounds produced by *S. cerevisiae*. Their biosynthesis is performed via amino acid catabolism, through the Ehrlich pathway [38]. Initially, transaminases, encoded by *aro*8, *aro*9, *bat*1, and *bat*2 genes, deaminate amino acids to the corresponding α-ketoacids. Secondly, α-ketoacids are converted into their corresponding aldehydes by enzymes encoded by one of the five decarboxylase genes (*Pdc1p, Pdc5p, Pdc6p, Aro10p*, and *Thi3p*) present in *S. cerevisiae*. Finally, the alcohol dehydrogenases catalyze the reduction of aldehydes into corresponding higher alcohols [39].

Recently, the selection of indigenous strains of *S. cerevisiae* has been widely applied, especially to improve the aroma profile and enhance the typicity of wines [40,41]. Inoculation of *S. cerevisiae* starters is a common practice, with dozens of different strains available on the market, selected for good fermentation kinetics (tolerance to osmotic and ethanol stresses and good nutritional efficiency in using available nitrogen sources). In addition, different starter cultures respond variably to the applied fermentation techniques and contribute differently to the development of sensory characteristics; as a result, yeast selection has become a key tool for developing and differentiating wines [42].

According to the International Organization of Vine and Wine (OIV), industrial starters can be marketed as: Active Dry Yeast (ADY), Active Frozen Yeast (AFY), Compressed Yeast (COY), Cream Yeast (CRY), Encapsulated (beads) or Immobilised Yeasts (ENY), and “*levain de tirage*” for sparkling wines [43].

The application of biotechnological tools, such as genetic engineering, in *S. cerevisiae* opens new possibilities for the development of novel, enhanced, or changed wine traits, qualities, tastes, aromas, or manufacturing methods [44]. In addition, more sophisticated genetic engineering techniques have gained increasing attention over the past ten years, mostly CRISPR/Cas9 (Clustered Regularly Interspaced Short Palindromic Repeats/CRISPR-associated protein 9), which uses the “adaptive immunity” mechanism naturally found in bacteria and archaea to construct a precise tool for genome editing [45,46]. In wine yeasts, this technique has been successfully applied to create engineered strains with lowered urea production [47], higher osmotic tolerance [48], or to improve wine aromatic complexity [46]. Moreover, a strain of *S. cerevisiae*, officially recognized by the Food and Drug Administration (FDA), has been engineered to perform malolactic fermentation [49].

### 2.2. Non-Saccharomyces Yeasts

NS yeasts commonly occur in freshly crushed grape must, and although in the past they were supposed to be spoiling agents, only in the last few years has their role been reevaluated [23,50]. As shown in Figure 4 [51], these yeasts are distributed among Saccharomycetales.

Nowadays, the interest in these unconventional native yeasts has greatly increased, leading large companies to market them as starters for industrial production [52]. Blends of *S. cerevisiae* and NS strains are also widely marketed and proposed to ensure both aromatic complexity and a proper fermentation process. An overview of the most widely used strains is given in Table 1.

In order to understand and characterize the most important technological features, over the years, the genome of non-*Saccharomyces* yeasts, such as the species *Torulaspora delbrueckii, S. pombe, Debaryomyces hansenii, Lachancea kluyveri, Lachancea thermotolerans, Millerozyma arinose, Candida glabrata*, *and Zygosaccharomyces rouxii*, has been fully sequenced [14]. More recently, the genome sequencing of *Starmerella bacillaris* type strains CBS 9494, FRI751, PAS13, PYCC 3044, and NP2 [53] revealed the important mechanism behind the high production of glycerol due to the presence of two copies of the *GPP1* tandem array gene [54]. Important genomic findings also highlighted the aromatic profile of wine linked to metabolic aspects of yeasts, such as the production of esters due to overexpression and low expression of the *EHT1* and *EEB1* genes in *T. delbrueckii* [55]; the reduction of ethyl acetate in *H. uvarum* due to the disruption of the *HuATF1* genes; and the role of the *ARO8* and *ARO9* genes in encoding aromatic amino acid aminotransferases in *H. vineae* [56,57,58].

## 3. Contribution of Selected Yeasts to Emerging Winemaking Demands

### 3.1. Enhancing the Aroma of Wine

The overall quality of a good wine is essentially determined by its aromatic and sensory profile, which is strongly linked to its typicity, origin region of production, and above all, by its ability to determine the consumer’s purchase choice [59].

In wines, more than 800 volatile organic compounds (VOCs) have been described and detected in concentrations ranging from ng/L to hundreds of g/L; however, only a few of them, at concentrations over their perception threshold, are odor-active compounds [60].

As recently reviewed by Romano et al. [57], the sensory complexity is mainly determined by three aromatic categories: primary aromas, characterized by terpenes, thiols, carotenoids, and norisoprenoids, originating from grape varieties that contribute significantly to wine typicity; secondary aromas, constituted mainly by higher alcohols, terpenoids, volatile fatty acids, esters, and phenols; carbonyl compounds and sulfur compounds, produced by yeasts during fermentation, which are involved in the “bouquet” of wine; and finally, tertiary aromas that are produced during wine aging (maturation and bottle aging), including volatile phenols, furanic compounds, and acetals. In particular, the main metabolisms involved in aromatic compound production are summarized in Figure 5.

Each of these compounds is able to influence, in different ways, the aromatic complexity of wine by imparting a specific aroma and/or off-flavor. In detail, higher alcohols can affect the final bouquet in both positive and negative ways; excessive amounts can produce a strong, pungent smell and taste, while appropriate levels yield fruity traits [58]. Volatile phenols, derived from the precursors of hydroxycinnamic acid present in grape must, can enhance the fragrance of wine by conferring the characteristic aroma of spices, smoke, and leather, as well as by enhancing the bitterness and astringency of wine; however, when their levels exceed the threshold, they can induce off-flavors such as “band-aid”, “barnyard”, or “stable” [61]. In this context, *Brettanomyces bruxellensis* is strongly involved in phenolic off-flavors, primarily linked to 4-vinylphenol and 4-ethylphenol from p-coumaric acid and to 4-vinylguaiacol and 4-ethylguaiacol from ferulic acid. These compounds confer undesired flavors, described as “horse sweat”, “rancid cheese”, or “musky” [62]. Esters, which are produced by yeast during fermentation, can expressively affect fruity flavors. The most widespread ester found in wine is ethyl acetate, which shows a fruity/solvent-like aroma at a threshold level of 150 mg/L. However, many other esters, including isoamyl acetate (banana aroma), isobutyl acetate (apple/pear aroma), ethyl hexanoate (fruity, strawberry, green apple, anise aroma), and 2-phenylethyl acetate (honey, fruity, flowery aroma), have been detected [63]. Moreover, volatile thiols are another class of aromatic compounds found in wine. They are mostly produced during fermentation and can confer either a pleasant or negative aroma, such as furfuryl thiol, identified in Bordeaux red wines (Petite Manseng variety), that exhibits roasted coffee flavors [64]. Among carbonyl compounds, acetaldehyde is the most commonly detected in wine, at concentrations ranging from 10 mg/L to 75 mg/L and with a sensory threshold value of 100 mg/L. At low concentrations, it can give a fruity flavor; however, it can also provide off-flavors like “bruised apple” and “nutty”, which are also used as indicators of oxidation in wine. Sulfur compounds are regarded as undesirable molecules, producing negative sensory attributes such as rotten egg smell [65], and even if they commonly occur at relatively low concentrations, they exhibit a very low detection threshold.

It is already well established that the microbial population present in vineyards and in musts, which includes *Saccharomyces* and NS yeasts, has a decisive influence on the final aroma complexity of wines and, consequently, on consumer preferences [66]. In this context, “microbial terroir” has been recognized as the pool of native yeasts that are geographically unique and imprint a specific and distinct olfactory fingerprint on wine [67,68]. Therefore, secondary aromas have a greater influence among the main volatile compounds that define the overall quality of wine, for which the yeast inoculum and the fermentation methodology play an essential role [69]. Despite this, the importance of indigenous isolates of *S. cerevisiae* is also amply demonstrated during the production of wine, reflecting defined territorial characteristics [70,71]. Several studies reported that the use of native *S. cerevisiae* yeasts leads to wines with higher ester concentrations and sweet/fruity aromas [72]. In particular, Agarbati et al. [73] demonstrated that the inoculation of different *S. cerevisiae* biotypes isolated from a winery that had never used commercial starters resulted in wines with different sensory characteristics compared to wines produced with commercial starters. In detail, wines fermented with autochthonous yeasts were characterized by the presence of noticeable esters such as isoamyl acetate (banana scent) and ethyl octanoate (apple, pear, and peach aroma). Consistently, Chen et al. [74] demonstrated how the use of indigenous *S. cerevisiae* yeasts, as opposed to industrial starters, can lead to an increase in the aromatic composition regarding the production of esters like ethyl hexanoate or ethyl acetate, thus conferring an aromatic distinctive imprint linked to the geographical location [74].

Among vineyard yeasts, the NS, when inoculated in must, produces a greater range of volatile metabolites compared to those obtained solely by *S. cerevisiae* [23]. In general, according to Carpena and co-workers [69] and as reported in Figure 6, three types of fermentations can be distinguished: spontaneous fermentations, in which selection of native population of NS and *S. cerevisiae* occurs, developing wines with high aroma complexity at the expense of less microbiological control; driven fermentations, in which inoculation with *S. cerevisiae* is performed inhibiting any spoilage growth or uncontrolled fermentation at the expense, however, of less aromatic complexity; and mixed fermentation, in which an initial inoculum with selected NS strains is usually performed to enhance the aromatic profile, followed by a sequential inoculum of selected *S. cerevisiae* starters to properly complete alcoholic fermentation; simultaneous inocula can also be performed by adding NS and *S. cerevisiae* at the same time.

It has already been ascertained how the presence of beta-glucosidase enzymes within NS yeasts is linked to the aromatic production in wines by releasing aromatic monoterpenes [75].

Strains of *Wickerhamomyces anomalus* have been described for their high beta-glucosidase activities; in particular, in Muscat Bailey A must, the co-fermentation with *W. anomalus*, followed by *S. cerevisiae*, results in a significant increase in linalool, citronellol, and geraniol contents. In a sensory evaluation, the flavor, taste, and overall preference scores of co-fermented wines were higher than those of the control [76]. Additionally, the aroma diversity and taste characteristics (intensity, astringency, complexity, and persistence) of Cabernet Sauvignon wines were improved through co-inoculation of five *W. anomalus* strains with *S. cerevisiae*, and three of them showed interesting results in terms of aroma complexity. In detail, two of them exhibited higher concentrations of esters (acetate esters, ethyl esters, and other esters), as 2-phenylethyl acetate, while a third strain showed the highest contents of titratable acidity, total phenolic, total tannin, and total anthocyanin, but also a higher level of norisoprenoids, as ionone and damascenone (rose and honey flavor, respectively), carbonyl compounds, as 2-octanone and nonanal (soap and herbaceous green flavor, respectively), higher alcohols, as iso-pentanol (whiskey, malt flavor), and esters [77].

Furthermore, beta-glucosidase activity has been detected in other autochthonous wine species, as in *Pichia fermentans,* recently investigated in simultaneous and sequential fermentations with a commercial starter of *S. cerevisiae,* both in synthetic and grape juice medium. Specifically, co-inoculation showed a higher production of terpenes and C_13_-norisoprenoids related to fruity and floral traits, exploiting glycosidase activity present in grape juice medium, probably starting with glycoside precursors present in grapes [78]. Within the *Pichia* genus, *Pichia kluyveri* is the only species to be commercially available, and its oenological potential is mainly related to its metabolism, which is able to release volatile compounds such as esters and thiols, especially 3-sulfanylhexan-1-ol acetate [3_SHA] (passionfruit, box tree aroma) [79], although beta-glucosidase seems to be poorly released by yeasts [80]. Moreover, a strain of *P. kluyveri* isolated from pear showed a good production of esters, such as ethyl-acetate (banana aroma), even though their concentrations were found to be higher than their threshold (200 mg/L), thus conferring off-flavors [81]. Likewise, in Riesling Italico grape must *P. kluyveri* yield the highest concentrations of total esters, glycerol, and i-valeric acid [82]. Therefore, when a strain is selected as a starter, its potential production of both SO_2_ and isovaleric acid (associated with rancid cheese off-flavor) must be considered [79].

Regarding *Torulaspora delbrueckii*, Bely et al. [83] demonstrated the capacity of lowering the acetic acid (which confers an undesirable “vinegary” trait) under standard conditions. In detail, in co-culture with *S. cerevisiae,* at a 20:1 ratio, a lower production of volatile acidity (53%), and acetaldehyde (60%), has been observed compared to fermentation with a pure culture of *S. cerevisiae*. According to those results, in Chardonnay and Palomino wines obtained with sequential inocula of an industrial strain of *T. delbrueckii*, lower levels of both acetic acid and acetaldehyde were obtained, although the wines resulted in enhanced aroma intensity of fresh and tropical fruit traits [84]. In Riesling Italico grape must inoculated with single cultures of different NS strains, *T. delbrueckii* showed a higher production of aromatic compounds, such as isoamyl acetate (banana), hexyl acetate (pear and apple), phenylethyl acetate (roses and honey), and ethyl decanoate (floral), which enhanced the aroma traits and the sensory traits of wines [85]. On the same grape cultivar, Dutraive et al. [82] performed different sequential inocula with different native strains, among which *T. delbrueckii* exhibited very high concentrations of higher alcohols and low acetic acid content. Furthermore, the production of glycerol, related to mouthfeel sensation [23], by *T. delbrueckii* was also evaluated in Cabernet Sauvignon wine. The aromatic profile was explored in the red wines obtained using indigenous and commercial NS yeasts in sequential fermentation with *S. cerevisiae* D254 starter. In detail, the fermentation performed with the autochthonous *T. delbrueckii* was characterized by a higher amount of glycerol of about 9.73 g/L after malolactic fermentation, whereas fermentation with a respective commercial strain (Prelude™ Chr. Hansen) resulted in a higher content of isoamyl acetate and a lower content of acetic acid [86]. However, glycerol production could be affected by many variables, such as grape cultivar and nitrogen composition [84].

Among notable non-conventional yeasts, *Lachancea thermotolerans* is widely used for its high lactic acid and low acetic acid production; hence, it can produce wine without the addition of lactic acid bacteria (LAB) for malolactic fermentation [87]. Furthermore, in *L. thermotolerans*, the Crabtree effect is much weaker than in *S. cerevisiae*, thus contributing to lower ethanol production due to aerobic respiratory metabolism [88]. Moreover, promising results were obtained in evaluating its biocontrol attitude [89]. In Merlot musts, sequential inocula with *L. thermotolerans* resulted in “fresher” wines with reduced acidity and ethanol and increased ester content, mainly ethyl acetate and ethyl lactate, thanks to the presence of lactic acid precursor [90]. Nevertheless, the efficiency of *L. thermotolerans* in consuming acetic acid can be highly inhibited by high concentrations of sugars (such as glucose) in musts under aerobically limited conditions [87]. Furthermore, scientific evidence reports divergent results about wine aromatic profiles using *L. thermotolerans*, which generally shows a low amount of aromatic compounds compared to *S. cerevisiae* [91].

Frequently used simultaneously with *L. thermotolerans*, *Schizosaccharomyces pombe* exhibits important characteristics in wines. *S. pombe,* being able to perform malo-alcoholic fermentation, can affect the pH of wine and cause microbiological instability, releasing ethanol and CO_2_. Therefore, the availability of *S. pombe* strains on the market is based on their ability to bio-deacidificate wines [92]. Generally, from an aromatic point of view, fermentation with *S. pombe* produces a wine with a weak aromatic complexity, a higher concentration of acetaldehyde, propanol, and 2,3-butanediol, and a lower ester content; this aspect is particularly appealing for wines in which the varietal aroma of grapes is preferred above the fermentative aroma [50]. However, when compared to wines fermented with *S. cerevisiae*, no discernible change in flavor has been observed [23]. However, in contrast to *L. thermotolerans*, one of the main negative aspects concerning *S. pombe* is the high levels of acetic acid production [50]. It is also important to point out that the use of mixed cultures of *L. thermotolerans* and *S. pombe* can improve both the structure and color of wine thanks to the production of flavylium ions and polysaccharides [93].

Focusing on the aromatic potential of native yeasts, *M. pulcherrima* was shown to have a substantial impact on fermentation due to its high release of aromatic compounds, such as varietal thiols, and low production of acetate, ethanol, and acids. However, during the early stages of fermentation, *M. pulcherrima* can deplete the nitrogen reserve at the expense of *S. cerevisiae*, and thus the potential for nitrogen deficits must be carefully considered to avoid sluggish fermentation [94]. Aromatic complexity attributable to *M. pulcherrima* can be related to its high beta-glucosidase activity [76]. Moreover, when the inoculum of *M. pulcherrima* and *Metschnikowia fructicola* on pre-fermentative cold maceration in Sangiovese must was evaluated through solid-phase extraction and GC/MS, results revealed higher content of some terpenes and C_13_-norisoprenoids, nerol (citric aroma), geraniol (geranium), 8-hydroxy-linalool (cis), 3-oxo-α ionol, and some esters, isoamyl lactate (cream, nut) and ethyl isoamyl succinate, than those found in both controls and wines treated with pectic enzyme [95]. Promising results were obtained by Escribano-Viana et al. [96], which reported an overall positive aroma profile with a consistent concentration of higher alcohols, including 2-phenylethanol (rose-scented), and by Binati et al. [97], which confirmed a promoted yield of esters and reduced volatile phenols. Using commercially available strains, it was also reported how the sequential inocula of the industrial *M. pulcherrima* AWRI1149 and AWRI305 strains, together with *S. cerevisiae,* produced Shiraz wine with increased intensity of several desirable aromas and lower off-flavors [98]. Similar results were reported recently by Naselli et al. [99], who examined the effect of *M. pulcherrima* inocula on the aroma and sensory complexity of Catarratto wines. Specifically, they realized a noticeably increased aromatic richness of the inoculated wines, which was also attributed to the presence of esters, the most prevalent of which was ethyl decanoate.

*Hanseniaspora*, specifically *H. guilliermondii*, *H. vinae*, and *H. uvarum*, are also relevant species from an enological point of view thanks to their high production of aromatic compounds, such as acetate esters, 2-phenylethyl acetate, and ethyl acetate, and their high enzymatic activities, including beta-glucosidase and beta-xylosidase [100]. In Ecolly and Cabernet Sauvignon wines produced by using *H. uvarum*, specific varietal and fermentation volatile compounds were found. Particularly, sequential inocula enhanced the aromatic profile of the final product by notably boosting fruity and flowery aromas. Nevertheless, excessive levels of *H. uvarum* affected the fermentation kinetics, producing wines with a nail-polish-like aroma due to higher levels of acetate esters and volatile phenolics [101]. Moreover, strains of *H. uvarum* may produce an excessive amount (up to 1 g/L) of acetic acid, imparting the typical vinegary smell [102].

Lately, the use of *Starmerella bacillaris* (synonym *Candida zemplina*) has recently become widespread. Generally, this yeast is capable of resisting high concentrations of ethanol, and thus it is commonly present in the intermediate/final stages of must fermentation [103]. The most important peculiarities of this species are its high glycerol production (up to 14 g/L) [17] and its fructophilic character [104]. The mixed culture of *S. bacillaris* and *S. cerevisiae* seems to enhance the concentration of fatty acids, esters, and higher alcohols compared to the pure culture of *S. cerevisiae* [105]. High amounts of monoterpenes, such as linalool, geraniol, ocimene, and nerol, were also detected in Sauvignon Blanc wines [106], although the presence of these compounds is highly grape variety-dependent [50].

### 3.2. Reduction of Ethanol Content

Global warming has critically affected wine production as it has allowed an acceleration in the ripening of grapes and, thus, a higher accumulation of sugars, resulting in higher alcoholic wines [107]. Alcohol consumption is certainly the cause of many health and social problems [108], and consumer attention to health issues has become increasingly pronounced [109]. In this context, the wine industry is increasingly oriented toward promoting wines with reduced ethanol content. Several strategies, including agronomic and fermentation techniques, have been developed; nevertheless, the use of selected yeasts might be a suitable alternative with reduced pressure on wine aroma complexity and quality [110]. In particular, it would be possible to produce wine with lower ethanol by inoculum of Crabtree-negative yeasts (NS yeasts), which would preferentially consume sugars by respiration rather than fermentation. This would allow producers to take advantage of the standardization of partial aeration of must [111].

Among the NS yeasts, *M. pulcherrima* appears promising in producing ethanol-reduced wines thanks to its aerobic respiratory metabolisms that, in suitable aeration conditions, can aerobically metabolize more than 40% of sugars, thus significantly reducing the ethanol yield [112]. For example, Contreras et al. [113] demonstrated that *M. pulcherrim*a, in sequential inoculum with *S. cerevisiae*, generated Shiraz wines with approximately 0.9% less ethanol than wines produced using *S. cerevisiae* in single cultures. A further study described the selection of a strain of *M. pulcherrima* for the production of lower-ethanol dry wines through a sequential inoculum with *S. cerevisiae*. In detail, based on the inoculation delay of *M. pulcherrima*, white wines (Chardonnay and Semillon blend must) contained between 0.6% and 1.2% less ethanol than wines fermented with *S. cerevisiae* in single culture [114]. Furthermore, a study carried out by Furlani et al. [115] described three strains of *M. pulcherrima* able to use more than 19 g/L of sugar to produce 1% (*v/v*) of ethanol in grape must. Additionally, *M. pulcherrima* and *Starmerella bombicola* are both promising wine yeasts to be used in immobilized forms in sequential fermentation to reduce ethanol content in Verdicchio must, with an ethanol reduction from 1.10 to 1.46% (*v/v*) and from 1.17 to 1.64% (*v*/*v*), respectively [116]. A recent study confirmed that a sequential fermentation with a native NS, *M. pulcherrima* CLI 68, and *S. cerevisiae* CLI 889 generated a reduction of alcohol content around 1.3% in Malvar wine with a higher glycerol content [117]. Furthermore, in Merlot grape juice, sequential inoculum of *M. pulcherrima* followed by *S. cerevisiae* resulted in about 11.7% *v*/*v* less alcohol production and a 12.3% reduction when observed using *Meyerozyma guilliermondii*. However, inoculation during fermentation with *M. pulcherrima* alone seems to negatively affect the overall aroma of the final product for producing ethyl acetate, which is lower if co-inoculated with *S. uvarum* [118]. Similar results were obtained by Rocker et al. [119] that showed how *M. pulcherrima* could be a promising yeast, reducing 3.8% ethanol in Riesling must at the expense of isovaleric acid.

It has also been shown that if *M. pulcherrima* can aerobically metabolize more than 40% of sugars, excessive aerobic conditions have an impact on ethyl acetate production. Canonico et al. [120] proved that, in Verdicchio must, the combined use of immobilized cells at a low level (5% (*w*/*v*) of beads) and at an aeration flow of 20 mL/L/min produced a relevant ethanol reduction (1.38% (*v*/*v*)), with the production of ethyl acetate still relevant but at an unacceptable level compared to those produced as free cells. Moreover, wines produced with *M. pulcherrima* and *Z. bailii* under 0.025 VVM (volume of air per volume of culture per minute) aeration resulted in a promising balance between ethanol reduction and volatile profile in Chardonnay must [121].

Moreover, an additional study using different species, namely *T. delbrueckii* and *Z. bailii*, showed the greatest potential for producing wines with reduced ethanol concentration under limited aerobic conditions and on defined media, with over 2% [*v*/*v*] reductions, depending on the aeration regime, compared to the anaerobic control [122].

Interesting results were also obtained using *Pichia* spp. In particular, *P. kluyveri* is able to produce about 0.36 g of ethanol per g of sugar, with a yield that is about 22% lower than that of *S. cerevisiae* [79]. Wines with reduced ethanol concentrations without excessive acetic acid levels were produced by sequentially inoculating *P. kluyveri* with *S. cerevisiae* in high-sugar Merlot grape musts [123].

However, *S. cerevisiae* must be used in conjunction with these NS strains since they usually exhibit a poor ability to drive the wine fermentation alone. Using only selected *S. cerevisiae* strains, Tronchoni et al. [124] reported low ethanol yield and less acetic acid production under aerobic conditions at laboratory scale, replacing NS yeasts as reported by Gonzalez et al. [111].

From an ethanol-reduction perspective, Gene Modification (GM) techniques have been applied to partially redirect carbon metabolism from ethanol production by shifting carbon to other endpoints [125]. In detail, the overexpression of *GPD1* and/or *GPD2* genes, encoding the glycerol-3-phosphate dehydrogenase isozymes, increases the synthesis of glycerol while decreasing the synthesis of ethanol [126]. In addition, Cuello et al. [127] developed a low-alcohol producer strain through a genetic modification of the *S. cerevisiae* Pdc2p transcription factor, encoded by the *PDC2* gene, that is involved in the regulation of the availability of the PDC (pyruvate decarboxylase) isozymes, which catalyze the reaction of pyruvate to acetaldehyde in the ethanol biosynthetic pathway. Through homologous recombination, the strain can reduce the ethanol concentration up to 1.89% *v*/*v* without affecting the overall fermentation kinetics. More recently, it was shown that by mutating the general transcription factor Spt15p, the TATA-binding protein, a strain of *S. cerevisiae* (YS59-409) was developed with an ethanol reduction of about 34.9% [128]. Nevertheless, only a few efforts have been made in this direction, probably because of consumer uncertainty about GM technologies.

### 3.3. Reduction of Sulfur Dioxide

Sulfur dioxide (SO_2_) is widely applied in winemaking at specific stages since it can inhibit the Maillard process and regulate oxidative activities, such as polyphenol oxidase, that affect the nutritional and sensory qualities of wine [129,130]. SO_2_ is mainly used for its antibacterial properties against spoilage microorganisms, to reduce the growth of molds during the initial stages of fermentation or deleterious bacteria and yeasts throughout fermentation, minimizing undesirable secondary fermentations. Furthermore, when SO_2_ is added before bottling, it can prolong shelf life and reduce the chance of undesirable aroma production [131]. On the other hand, sulphite exposure has been related to bronchoconstriction, urticaria, headaches, dermatitis, diarrhea, or worsening of asthmatic symptoms [132,133]. Therefore, it is necessary to take into account the long-term effect on consumers. In Europe, the International Organization of Vine and Wine (OIV) recommends the following limit for SO_2_ to be added during winemaking: 150 mg/L for red wines, 200 mg/L for white wines, 300 mg/L for wines with a sugar content over 4 g/L, and 400 mg/L for sweet and special wines [134].

Bioprotection has been widely proposed as an effective approach to replacing SO_2_ in winemaking [135]. Specifically, selected yeast strains can inhibit the growth of unwanted microorganisms since they can overtake spoilage fungi for space and nutrients as well as synthesize biocontrol compounds as killer toxins [136]. Among wine spoilage microorganisms prevented by SO_2_, *Brettanomyces bruxellensis* is the most important since, as reported before, its presence is related to undesirable traits such as the typical “horse sweat” odor [62,137].

In this context, the use of NS yeasts as biocontrol has already been established, in particular by employing them both at harvest and during fermentation, limiting the development of spoilage microorganisms and preventing chemical and enzymatic oxidation [102].

The NS *Torulaspora delbrueckii* and *M. pulcherrima* species have been widely used to drive industrial fermentations without reduced-added sulfites. The ability of *T. delbrueckii* to bioprotect is attributable to the killer toxin TdKT, which exhibits a broad spectrum against wine spoilage yeasts without altering *S. cerevisiae* growth [138]. Windholtz et al. [139] showed that *T. delbrueckii* was found to have efficacy in bioprotection under sulfite-free conditions without altering the complex flavor profile of Merlot red wines. Similar results were obtained by Simonin et al. [140], although native yeasts are still less effective than sulphites at preventing oxidation of the must.

On the other hand, the antibacterial activity of *M. pulcherrima* seems to be related to the pulcherriminic acid (precursor of pulcherrimin), which immobilizes the iron in the growth medium; in this case, it has already been proven that *M. pulcherrima* is efficacious in inhibiting *B. bruxellensis* [141] as well as fungal species, such as *Botrytis cinerea* [142]. A recent study verified that, under experimental conditions, *M. pulcherrima* reduced the growth of spoilage microbiota in Pinot Noir grapes in pre-fermentation stages without any change in aroma profile [135]. However, due to the amensalism of *M. pulcherrima* to *S. cerevisiae* through iron depletion, adverse effects on fermentation kinetics could be observed when mixed cultures are used [143].

Among the non-conventional yeasts, the use of *W. anomalus* against *Brettanomyces* additionally stands out due to the presence of the killer toxin KTCf20 in some specific strains [144]. Similarly, Comitini et al. [145] demonstrated the efficiency of selected strains of *W. anomalus*, isolated from dairy environments, as a biocontrol agent against *B. bruxellensis.*

Furthermore, strains of *Candida pyralidae* showed production of two killer toxins, CpKT1 and CpKT2, capable of suppressing the proliferation of *B. bruxellensis* in similar winemaking conditions [146].

Likewise, in *S. cerevisiae*, the presence of a naturally synthesized biocide, saccharomycin, has recently been detected; in particular, it was evaluated that the *S. cerevisiae* CCMI 885, both in natural and synthetic must, prevented the growth of *B. bruxellensis* by greatly reducing SO_2_ addition [147]. Using selected strains of *S. cerevisiae*, Capece et al. [148] proposed a biotechnological approach to perform wine fermentation without SO_2_ addition, demonstrating the dominance of starter cultures over indigenous microbiota. In terms of sensory complexity, as previously reported by Miceli et al. [149], wines without SO_2_ (with or without bioprotection) were recently described as notably different from wines with SO_2,_ mainly after 2 years of aging [150]. However, in the literature, scarce and divergent findings are reported on the sulphites’ impacts on the aromatic profile [135].

However, despite the encouraging findings, the preservation efficacy of SO_2_ in wines remains relevant, given that no other approach has been suggested as being able to totally replace SO_2_ [151]. The output of sulphite released by yeasts should also be considered since *S. cerevisiae* can produce an average of about 10 mg/L of SO_2_, which increases to 30 mg/L for some selected strains [152].

## 4. Enological Yeasts and Health-Related Compounds

### 4.1. Biogenic Amine Reduction Content

Biogenic amines (BAs) are low-molecular-weight compounds obtained from amino acid precursors, mainly by microbial decarboxylation. Several investigations have reported the presence of different BAs in any type of wine, such as histamine, tyramine, and putrescine [153]. BA levels in wine depend on many factors, including grape variety, climatic conditions, agricultural practices, vinification techniques, strains used in fermentation and aging, and wine parameters such as pH, alcohol content, and sulfur dioxide content [154]. Red wines showed higher levels of BAs, especially putrescine (PUT) and histamine (HIS), than white wines, with an average concentration of 7.30 and 2.45 mg/L, respectively [155]. It is relevant to highlight that, in sensitive consumers, exposure to high BA concentrations may result in health issues such as headache, anaphylaxis, nausea, hypertension, nervous symptoms, organ failure, and renal intoxication. Furthermore, ingestion of high levels of HIS is associated with the onset of histamine poisoning, characterized by hypertension, flushing, tachycardia, and gastrointestinal symptoms; ingestion of tyramine is associated with hypertensive crisis and cardiovascular symptoms [156,157]. Lactic acid bacteria (LAB) are considered the main cause of BA production, either during malolactic fermentation or due to contamination during various stages of winemaking [158]. Among wine yeasts, *B. bruxellensis* shows the highest amine production, while it appears to be a strain-dependent condition for *S. cerevisiae* [159]. Moreover, it has already been reported that some non-conventional yeasts, such as *Debaryomyces hansenii*, can reduce the BA contents in food even though the species is not suitable for winemaking due to its reduced ethanol tolerance and production of unpleasant aromas [160].

Therefore, exploring other NS yeasts that do not produce BAs as starters can reduce the concentrations of these compounds in wine. In this way, *L. thermotolerans* and *Shizosaccharomyces pombe* can be combined to simultaneously perform AF and malic acid degradation, thereby eliminating MLF in the traditional winemaking process and reducing the activity of LAB and, subsequently, the risk of BA production in wine [161,162,163]. Furthermore, it was similarly observed that the co-inoculum of *T. delbrueckii* TD20 with *H. vineae* HV6 significantly reduced tyramine level in Petit Manseng must [164], and it has been reported that the use of the *H. uvarum* FS35 strain, in sequential inoculum with commercial *S. cerevisiae*, produced a wine with lower level of putrescine and higher amounts of glycerol, lactic acid, acetic acid, phenylethyl alcohol, ethyl acetate and beta-phenylethyl acetate compared with the control fermentation; proving that *H. uvarum* FS35 strain is a promising strain to reduce BAs in wines through copper amine oxidase 1 (CuAO1) activity that lead to a coordinated response in the oxidative deamination of putrescine to 4-amino-butanal and subsequent dehydrogenation to 4-amino-butanoate [165].

### 4.2. Resveratrol-Increasing Content

Resveratrol (3,5,4′-trihydroxy-trans-stilbene) belongs to the polyphenols’ stilbenoids group, possessing two phenol rings linked by an ethylene bridge. This natural polyphenol has been detected in more than 70 plant species, especially in grapes’ skin and seeds, and was found in discrete amounts in red wines and various foods [166]. Resveratrol has been associated with a wide range of pharmacological properties such as anti-inflammatory, anti-oxidative, anti-aging, anti-diabetic, cardioprotective, and neuroprotective attributed to its anti-oxidative, anti-inflammatory, and immuno-modulating effects [167]. Furthermore, the antimicrobial activity of this compound has been widely demonstrated [168]. However, due to the alcohol content, these beneficial effects cannot be achieved exclusively by consuming of wine [169]. In this context, finding biotechnological solutions for increasing the availability of resveratrol in wines could be necessary to achieve a healthy effect. The average concentration of total resveratrol in red wine is 7 mg/L, 2 mg/L in rosé wines, and 0.5 mg/L in white wine, but it ranges widely according to grape variety, geographical indication, and winemaking processes [170]. Despite the agronomic practices that can be adopted to increase resveratrol availability, the use of selected yeasts during winemaking could be a suitable alternative, especially those producing beta-glucosidase enzymes [171].

The use of genetically modified *S. cerevisiae* strains to increase resveratrol levels has already been amply demonstrated in previous years [172,173,174,175,176]. For instance, the industrial strain *S. cerevisiae* EC1118 could in vitro produce 8.249 and 3.317 mg/L of *trans-*resveratrol with or without antibiotic addition, respectively, by incorporating the coenzyme A ligase *4cl* gene, lacking in *S. cerevisiae* and coming from *Arabidopsis thaliana,* and the resveratrol synthase gene from *Vitis vinifera* [177].

Alternatively, the use of selected native yeast strains is also a viable alternative; Gaensly et al. [178], isolated beta-glucosidase-producing strains of *H. uvarum* capable of increasing resveratrol content up to 102% during fermentation of Bordò must (*V. labrusca*) through hydrolysis of piceides. Lately, a study analyzed that the *trans-*resveratrol concentration in red Tempranillo wines inoculated with *T. delbrueckii* was three times higher than in wine obtained by using *S. cerevisiae* and 4.5 times higher when *Z. bailii* was inoculated [179,180]. Nevertheless, more and more studies increasingly focus on the reuse of wine wastes for bioactive compound recovery, such as resveratrol; in this perspective, Costa et al. [180] showed that the use of an engineered strain of *S. cerevisiae*, which is also capable of hydrolyzing xylose, on wine waste, such as musts, pruning residues, and wine lees, can lead to modest resveratrol production, representing a viable sustainable alternative in a circular economy context.

### 4.3. Probiotic Wine

Probiotics are dietary supplements that include viable microorganisms that can survive in (or briefly colonize) the human gastrointestinal tract and have a positive impact on consumer health. Exogenously added probiotics of a species or strain can temporarily colonize the intestinal tract and re-establish the microbiota when the normal native microbiota has been disrupted, restoring essential physiological functions [181].

Among *Saccharomyces* yeasts, *Saccharomyces boulardii* is the only species officially recognized to have significant probiotic effects on human health because of its immune-modulatory, antioxidant, antiviral, antibacterial, anti-inflammatory, and anticarcinogenic properties [182]. The advantageous effects of *S. boulardii* as a probiotic are supported by observable phenotypic features and physiological characteristics, including viability at low pH, ideal growth temperature, and tolerance to the gastrointestinal environment [183]. Furthermore, since it is naturally resistant to antibiotics, yeast cannot contribute to the spread of antimicrobial resistance [184]. Through investigations of synthetic must and in vitro testing, De Paula et al. [185] validated *S. boulardii* as potentially able to tolerate ethanol stress, offering new possibilities for the formulation of novel probiotic wines. Recently, the use of *S. boulardii* as a probiotic in winemaking has been reported for the first time [186] using the strain *S. cerevisiae* var. *boulardii*, CECT 1474, in the fermentation of a *Monstrell* must for the production of rosé wine; the results showed how the final product exhibited no significant differences from an aromatic point of view, compared to the control, and also preserved probiotic characteristics for about 6 months even during storage at 25 ± 0.5 °C and ~4 °C. On the other hand, among NS yeasts, several strains, isolated from different locations, with potential probiotic and healthy activities have been reported, especially belonging to the genera *Debaryomyces*, *Kluyveromyces*, *Yarrowia*, and *Torulaspora*, but also *Candida*, *Pichia*, *Hanseniaspor*a, *Metschnikowia*, and species such as *L. thermotolerans* and *Metschnikowia ziziphicola* [187,188]. Within native yeasts, *Kluyveromyces marxianus fragilis* B0399 was the first to be reported as a probiotic, officially recognized to be used in both livestock feed and human consumption [189,190] thanks to its strong adherence to human enterocyte-like Caco-2 cells and its ability to modulate the immune response by inducing pro-inflammatory cytokines in peripheral blood mononuclear cells [190].

Probiotic characteristics have also been recently detected in other NS yeasts, and in a recent study, 15 strains belonging to the genus *Pichia* were selected for their potential probiotic activities like adhesion and resistance to gastrointestinal conditions, showing promising results compared to *S. cerevisiae* var. *boulardii* CNCM I-745 used as [191]. In Table 1 an overview on the non-*Saccharomyces* yeasts most commonly used in winemaking process, and their positive and negative effects on aroma compounds and biotechnological impacts is reported.

**Table 1 microorganisms-11-01338-t001:** Main non-*Saccharomyces* yeasts with promising effects on aroma and the winemaking process. Their availability on the market, from the main industrial starter companies, is also reported. AFY (active frozen yeast); ADY (active dry yeast); CRY (cream yeast); FLY (fresh liquid yeast); ENY (encapsulated yeast).

Yeast (Teleomorphic/ Anamorphic Form)	Aroma and Quality Contribution	Main Aromatic Metabolites	Flavors [57]	Biotechnological Impacts	Negative Impacts	Producer [52,58]	Formula	References
*Pichia* *kluvveri*	High production of thiols and ester	3-sulfanylhexan-1-ol, acetate	Passionfruit, box tree	Low ethanol yield Potential probiotic activities	Isovaleric acid and H_2_S production	CHR Hansen	AFY	[66,67,68,69,70,175]
*Hanseniaspora* spp./*Klockera* spp.	High production of esters High production of β-glucosidase	2-phenylethyl- acetate Ethyl acetate	Floral, rose Fruity	Potential resveratrol increasing Biogenic amine reducing	Acetic acid production in *H. uvarum*	Oenobrands (*H. vinae*)	ADY	[88,89,90,152,166]
*Torulaspora* *delbruekii/Candida colliculosa*	Low acetic acid High glycerol production High production of esters	Isoamyl acetate Hexyl acetate Phenylethyl acetate	Banana, pear Pear, apple Rose, honey	Potential use as biocontrol in SO_2_-reduced wines	Oxidation not fully controlled as sulphite addition	Lallemand CHR Hansen Laffort Agrovin Enartis Oeno Probiotec	ADY AFY ADY ADY ADY/ CRY ADY FLY	[70,71,72,73,126,127,128]
*Lachancea* *thermotolerans*	High Lactic acid production Low acetic acid production Improvement of color and structure	Ethyl acetate Ethyl lactate	Fruity Fruity, buttery	Low ethanol yield Potential biogenic amine reducing in co-inoculation with *S. pombe* Biocontrol agent	Possible lack of aromatic complexity	Lallemand CHR Hansen Enartis Lamothe-Abiet AEB Group Probiotec	ADY AFY CRY ADY ADY FLY	[75,76,77,78,79,149]
*Metschnikowia* *pulcherrima/Candida pulcherrima*	High production of β -glucosidase High productions of thiols	2- phenylethanol	Rose	Low ethanol yield Potential use as biocontrol in SO_2_-reduced wines	High ethyl- acetate Amensalism over *S. cerevisiae*	AEB group Lallemand	ADY ADY	[64,82,83,84,100,129]
*Starmerella* *bacillaris/Candida zemplinina*	Fructophilic character High glycerol production Terpenes production	Linalool Geraniol	Rose Rose	Low ethanol yield High resistance to ethanol	-	BioEnologia	CRY	[16,91,92,93,94,104]
*Wickerhamomyces anomalus/Candida pelliculosa*	High production of β-glucosidase High productions of acetate esters	2-phenylethyl acetate Isoamyl acetate Ethyl acetate	Floral, rose Banana, pear Fruity	Potential use as biocontrol in SO_2_-reduced wines	-	Probiotec	FLY	[13,64,132]
*Schizosaccharomyces pombe*	Malic acid reduction Improvement of color and structure	Propanol 2,3-butanediol	Pungent, harsh Fruity, butter	Potential biogenic amine reducing in co-inoculation with *L. thermotolerans*	Acetic acid production	Proenol BioEnologia	ENY CRY	[50,80,81,150]

## 5. Winemaking Sustainability and the Circular Economy

Climate change appears to pose more and more of a threat every year, which is why sustainable and environmentally friendly approaches must be adopted in industrial production. As has already been amply demonstrated, the wine sector, especially in Italy and generally in Europe, represents an unparalleled economic source given the high quantities of wine produced. As a result, the wine industry yields a considerable amount of waste products (like grape pomace and lees) but also chemicals derived from enological practices (sulphites), which emerge as environmentally unsustainable. Furthermore, energy costs resulting from the winemaking process must also be considered.

Vinification wastes contribute to ecological issues since the use of fermentative wastes, combined with other substances, poses a threat to both the environment and public health. Accordingly, it is crucial to take ecological precautions to safeguard the environment, and even more so considering the financial benefits of retrieving vineyard waste [192]. Considering the broad aspects described in this review, the development and use of commercial non-*Saccharomyces* yeast starters by companies appear promising for making wine in a more sustainable manner and in a circular economy context.

The use of specific yeast strains at different stages of the production process (pre- and post-fermentative stages) could represent an opportunity to improve energy savings, cut costs, and bring companies in a greener direction. The use of selected strains to ferment at desired temperatures could be a strategy to reduce heat dissipation and, therefore, energy saving [193]. Additionally, non-*Saccharomyces* yeasts can also be used as bio-protection in wines, thus avoiding the possible occurrence of spoilage and stuck fermentations (which would create further economic losses) and reducing the dispersion of chemical substances such as SO_2_ [92].

Regarding the treatment and management of vineyard waste, introducing the knowledge of biorefinery might have great outcomes regarding the sustainability of the environment as well as creating extra economic income [194,195]. Bioconversion can be employed to provide grape seed oil, ethanol, condensed tannins, tartaric acid, phenolic-rich extracts, bacterial cellulose, succinic acid, and, most importantly, bioactive compounds using grape pomace, stalks, and wine lees; in this way, a full valorization of winery wastes in a bioeconomy framework can be performed [196].

Grape pomace (GP) might be a valuable source of useful bioactive substances such as antioxidants, nutraceuticals, single-cell proteins, and volatile chemical compounds, with an emerging research interest in their therapeutic effects on human and animal health and their use as feed additives in livestock [197].

Wine lees (WL) are common winery wastes generated after the fermentation process at the bottom of wine containers, mainly composed of precipitated yeast, whose recovery has been widely reported in the literature in recent years [198], in particular for the extraction of bioactive compounds such as mannoproteins [199] and β-glucan [200], but also for the isolation of non-*Saccharomyces* species with remarkable antioxidant properties [201].

Alternatively, waste from the wine industry can be used as a substrate by oenological yeasts for the production of useful compounds, as reported by Williams et al. [202] regarding the use of *Kluyveromyces marxianus* strain Y885 for secreting pectinolytic enzyme. Thus, the data collected to date, already confirmed at the laboratory level, must be validated on a larger scale in order to establish ways to exploit these species at the industrial level. As a result, more research may lead to new perspectives and fascinating advances in the field of oenological sustainability.

## 6. Conclusions and Future Perspectives

The wine sector represents one of the most relevant market segments within the agribusiness economy; therefore, the challenges involved are highly variable and related to consumer requests. In the last century, the prominent role of *S. cerevisiae* in winemaking has been widely demonstrated and supported by producers. Furthermore, in recent years, “green” issues have become pivotal themes, and consumers are increasingly directed to consume safer products with a guaranteed provenance. Nowadays, with the introduction of advanced technologies, the revaluation of NS species has been greatly considered. The exploitation of native yeasts, isolated and selected directly from the production area, not only contributes to the qualitative and aromatic identity of a region but also helps in producing wines with reduced ethanol, SO_2_, and other substances that could compromise human health. Furthermore, the reduction of chemicals during the various stages of winemaking could be completely or partially replaced by a suitable and sustainable microbial approach in an environmentally friendly way. Moreover, industrial starter cultures can be set up by mixing selected strains exhibiting the most promising technological traits to obtain enhanced wines.

## Figures and Tables

**Figure 1 microorganisms-11-01338-f001:**
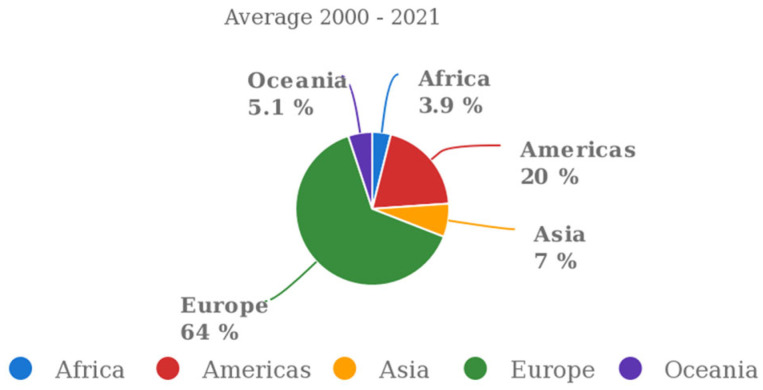
Production share of wine by region [source: https://www.fao.org/faostat/en/, accessed on 18 May 2023].

**Figure 2 microorganisms-11-01338-f002:**
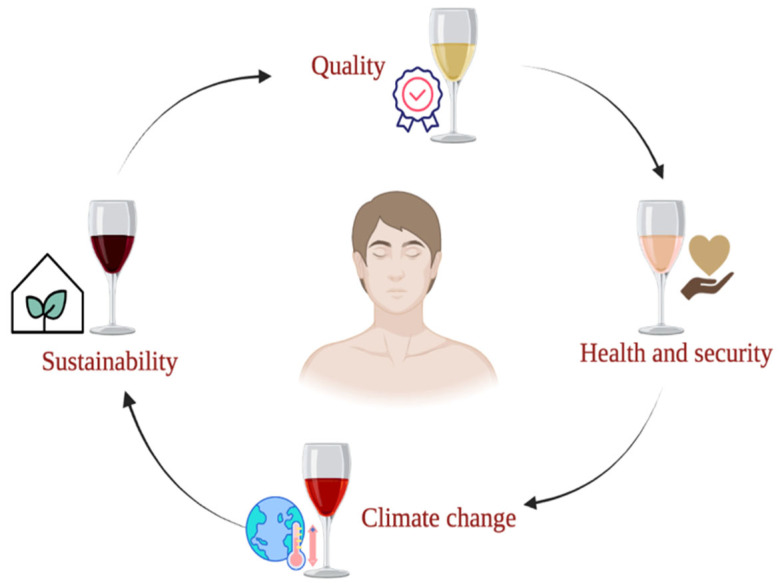
Perception needs of current wine consumers.

**Figure 3 microorganisms-11-01338-f003:**
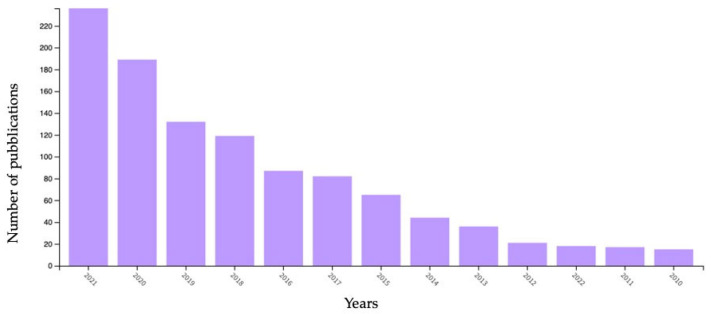
Records for “wine sustainability” found in the Web of Science between 2010 and 2021.

**Figure 4 microorganisms-11-01338-f004:**
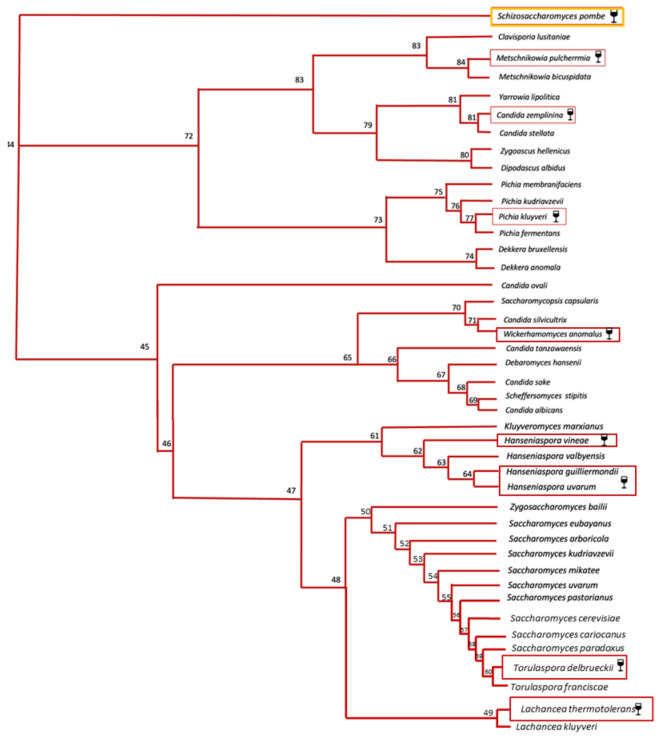
Phylogenetic tree showing the 41 yeast species of Saccharomycetales, obtained on the basis of the 18S ribosomal DNA sequence. Adapted from [51]. *Schizosaccharomyces pombe* (belonging to Schizosaccharomycetales) is represented as the outgroup species. Among the displayed species, the yeast species in the red boxes represent the NS yeasts of oenological interest described in the present review.

**Figure 5 microorganisms-11-01338-f005:**
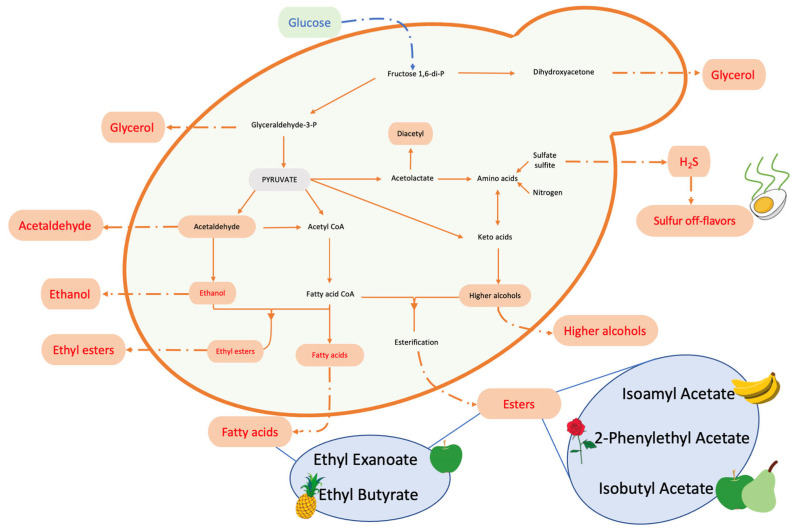
Main metabolic pathways in secondary aroma production performed by yeasts.

**Figure 6 microorganisms-11-01338-f006:**
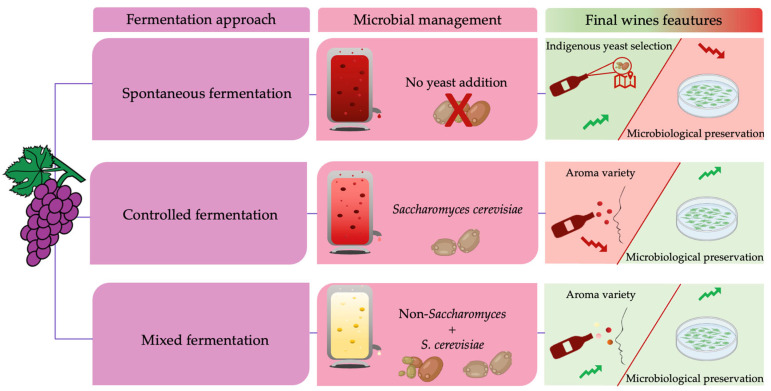
Wine features resulted from applying different fermentation approaches.

## Data Availability

No new data were created or analyzed in this study. Data sharing is not applicable to this article.

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
