# Peer review of "Inside Current Winemaking Challenges: Exploiting the Potential of Conventional and Unconventional Yeasts"

_microorganisms, 2023, doi:10.3390/microorganisms11051338_

Round 1

Reviewer 1 Report

The present study mainly reviews the effects of traditional and non-traditional yeast on the aroma of wine, the ability to reduce ethanol and sulfur dioxide content, and the benefits to human health. It shows that yeast is a valuable source of nutritional value in wine. It also indicates that yeast can be modified biologically for more excellent characteristics. The current research provided some new data and interest results, and the manuscript was well organized, but some issues still need to be clarified further according to the following comments.

1.       The last paragraph of the introduction indicated that in the face of consumer demand for wine safety and other aspects, the use of selected yeast and NS yeast strains in the production process could meet these requirements of consumers. However, lines 84 and 85 described that the existing wine-making technology is safer, and the addition of NS yeast will increase the complexity of wine composition, which is not consistent with the conclusion. It is suggested to provide the relative content to support the safety of wine.

2.       3.1, lines 283: The microbial population in the vineyard and wine (including yeast and NS yeast) has a decisive influence on the final aroma complexity of the wine. However, the influence of traditional yeast on the aroma is not discussed in the following paragraph.

3.       In line 32, customers must be “custom”;

4.       In line 296, Cane must be “can”;

5.       The sub-title of  4.2. Resveratrol “ is  ambiguous and should be further clarified as “4.1 Biogenic amine reduction content. “

Author Response

Response to Reviewer 1

  1. The last paragraph of the introduction indicated that in the face of consumer demand for wine safety and other aspects, the use of selected yeast and NS yeast strains in the production process could meet these requirements of consumers. However, lines 84 and 85 described that the existing wine-making technology is safer, and the addition of NS yeast will increase the complexity of wine composition, which is not consistent with the conclusion. It is suggested to provide the relative content to support the safety of wine.

R1. Thank you for the comment. Existing winemaking techniques (carried out with traditional S. cerevisiae starter cultures) are to be understood as safe from a microbiological point of view since, according to scientific literature, they guarantee proper fermentation without the occurrence of undesirable fermentations. On the other hand, the use of non-Saccharomyces yeasts can be considered safe from a health point of view since they can benefit the final product by guaranteeing a reduced ethanol content, a possible reduction in toxic compounds and an increase in important antioxidants such as resveratrol. We have reviewed the sentence in the revised manuscript (lines 95-105)

  1. 3.1, lines 283: The microbial population in the vineyard and wine (including yeast and NS yeast) has a decisive influence on the final aroma complexity of the wine. However, the influence of traditional yeast on the aroma is not discussed in the following paragraph.

R2. Thank you for the comment. As with non-Saccharomyces yeasts, we also introduced the importance of using native S. cerevisiae yeasts for improving the aromatic profile and valorising the terroir (lines 422-436).

  1. In line 32, customers must be “custom”;

R3. Thank you for the comment. We modified the typing error in the revised manuscript.

  1. In line 296, Cane must be “can”;

R4. Thank you for the comment. We modified the typing error in the revised manuscript.

  1. The sub-title of  “4.2. Resveratrol “ is  ambiguous and should be further clarified as “4.1 Biogenic amine reduction content. “

R5. Thank you for the comment. We modified the sub-title in the revised manuscript.

Reviewer 2 Report

The abstract contains main required components, and it forms coherent text with logical conclusions and interactions between its immanent parts.

The title of the paper is well formulated and it covers the content. The introduction logically follows the aim of the paper, and it provides valuable introspection into an unsolved topic.

Methodological part of the paper is suits current scientific standards. The results are presented clearly. The interpretation of tables and figures is acceptable. However, wider context of the presentation of the results should be applied to make the results really understandable for the audience. The level of the author’s knowledge is satisfying. 

Here few comments to improve the manuscript's quality:

- It would be useful to add/expand the paper's section in which authors explains how their results can be exploited by winemaking companies and policymakers

- It would be useful to add/expand the discussion on climate changes (listed in the introduction and never discussed in deep) and the role of yeast/yeast management connected with winemaking practices.

Author Response

Response to Reviewer 2

  1. It would be useful to add/expand the paper's section in which authors explains how their results can be exploited by winemaking companies and policymakers

R1. Thank you for the comment. In the light of the important results obtained so far and described in this review, we strongly believe that new commercial starters based on non-Saccharomyces yeast strains may represent an important opportunity for companies to operate in a more environmentally sustainable manner, in an increasingly compromised climate context. We have expanded this part in section 5 in the revised manuscript (Lines 1025-1072).

  1. It would be useful to add/expand the discussion on climate changes (listed in the introduction and never discussed in deep) and the role of yeast/yeast management connected with winemaking practices.

R2. Thank you for the comment. We have described the possible benefits of using selected yeast strains, to help producers fit into a sustainable context, in section 5 in the revised manuscript (lines 1025-1072).

Reviewer 3 Report

This survey details the use of NS yeasts in winemaking. The contents are thorough with enough references but honestly said, I find other review papers more informative. Please check, for example,

Masneuf-Pomarede et al. Frontiers in Microbiology doi: 10.3389/fmicb.2015.01563

Jolly et al. FEMS Yeast Res DOI: 10.1111/1567-1364.12111

Major points:

* Please add a figure of 18S ribosomal phylogeny for NS yeasts in this review. Their relationship with Saccharomyces is important (some were previously grouped in Saccharomyces). In writing their names, teleomorphic/anamorphic forms need consideration. For example, Pichia hansenii is a synonym and it should be called Debaryomyces hansenii. Please add these synonyms in Table 1.

* Add some information on genome availability and metabolic characters of NS yeasts. Not in great details but genome features are essential these days. This makes a good "challenge" as in the title.

Minor points:

* English:  Needs a proofreading for typos. For example, L 296 cane -> can; L.666 specie -> species; and so on.

* Section titles: please reconsider. I do not think that Section 2 discusses ecology, and Section 4 does not really discuss health. Wine metabolites only.

Author Response

Response to Reviewer 3

  1. Please add a figure of 18S ribosomal phylogeny for NS yeasts in this review. Their relationship with Saccharomyces is important (some were previously grouped in Saccharomyces). In writing their names, teleomorphic/anamorphic forms need consideration. For example, Pichia hansenii is a synonym and it should be called Debaryomyces hansenii. Please add these synonyms in Table 1.

R1. Thank you for the comment. We added a figure concerning 18S ribosomal phylogeny of NS and the correct nomenclature in table 1 in the revised manuscript.

  1. Add some information on genome availability and metabolic characters of NS yeasts. Not in great details but genome features are essential these days. This makes a good "challenge" as in the title.

R2. Thank you for the comment. We have expanded the current availability of the genome of the non-Saccharomyces yeast and pointed out some of the important metabolic aspects concerning technological features. (Lines 297-309)

  1. English:  Needs a proofreading for typos. For example, L 296 cane -> can; L.666 specie -> species; and so on.

R3. Thank you for the comment. We corrected the typing errors in the revised manuscript.

  1. Section titles: please reconsider. I do not think that Section 2 discusses ecology, and Section 4 does not really discuss health. Wine metabolites only.

R4. Thank you for the comment. We reformulated the titles in the revised manuscript in order to be more in line with the content.

Round 2

Reviewer 2 Report

I am currently fine with the current manuscript version. The authors have properly addressed my comments.

Author Response

Dear authors, some improvements are still needed for your review as listed below

Many wrong wording examples, here are some catched here and there: Line 127: fermentating Line 128: presents Line 217: can be market Line 335: isomyl Please, re-check the whole manuscript for this kind of mistakes and please, check the wording, e.g. Line 25.

R1. Thanks for the comment. We checked the manuscript and corrected the typing errors in the revised version.

Lines 101-110: this part should be shortened and clarified. Just restrict to the right searching words for your target.

R2. Thanks for the comment. We rephrased the sentence in the revised manuscript (lines 100-107).

Section 2.1: I think you should recall briefly, here or at line 64, the role and central metabolism of S. cerevisiae, not give it for granted

R3. Thanks for the comment. We have recalled briefly the principal aspect of S. cerevisiae metabolism which differs from that of non-Saccharomyces (lines 285-291).
